# Recent Advances in g-C₃N₄ Photocatalysts: A Review of Reaction Parameters, Structure Design and Exfoliation Methods

**Junxiang Pei, Haofeng Li, Songlin Zhuang, Dawei Zhang and Dechao Yu ***

Engineering Research Center of Optical Instrument and System, Ministry of Education and Shanghai Key Laboratory of Modern Optical System, University of Shanghai for Science and Technology, Shanghai 200093, China; jxpei@usst.edu.cn (J.P.); lhf09042023@163.com (H.L.); slzhuangx@aliyun.com (S.Z.); dwzhang@usst.edu.cn (D.Z.)

*** Correspondence: d.yu@usst.edu.cn**

**Abstract:** Graphitized carbon nitride (g-C₃N₄), as a metal-free, visible-light-responsive photocatalyst, has a very broad application prospect in the fields of solar energy conversion and environmental remediation. The g-C₃N₄ photocatalyst owns a series of conspicuous characteristics, such as very suitable band structure, strong physicochemical stability, abundant reserves, low cost, etc. Research on the g-C₃N₄ or g-C₃N₄-based photocatalysts for real applications has become a competitive hot topic and a frontier area with thousands of publications over the past 17 years. In this paper, we carefully reviewed the recent advances in the synthesis and structural design of g-C₃N₄ materials for efficient photocatalysts. First, the crucial synthesis parameters of g-C₃N₄ were fully discussed, including the categories of g-C₃N₄ precursors, reaction temperature, reaction atmosphere and reaction duration. Second, the construction approaches of various nanostructures were surveyed in detail, such as hard and soft template, supramolecular preorganization and template-free approaches. Third, the characteristics of different exfoliation methods were compared and summarized. At the end, the problems of g-C₃N₄ materials in photocatalysis and the prospect of further development were disclosed and proposed to provide some key guidance for designing more efficient and applicable g-C₃N₄ or g-C₃N₄-based photocatalysts.

**Keywords:** photocatalyst; g-C₃N₄; reaction parameters; structure design; exfoliation



## 1. Introduction

In order to alleviate the problem of global warming and energy shortage, the development of renewable energy has become a major practical problem to be urgently solved by researchers all over the world [1]. As the most important renewable energy on the Earth, solar energy can be said to be inexhaustible. But so far, limited by various energy conversion technologies, the development and utilization of solar energy in the field of photocatalysis is far from enough. Recently, graphitic carbon nitride (g-C₃N₄) has attracted extremely wide attentions in photocatalysis due to its special band structure, stable properties, low price, and easy preparation [2–6]. The g-C₃N₄ is comprised of only carbon and nitrogen elements, which are very abundant on the Earth. Importantly, the g-C₃N₄ materials can be easily fabricated by thermal polymerization of abundant nitrogen-rich precursors such as melamine [7–16], dicyandiamide [17–22], cyanamide [23–25], urea [18,26,27], thiourea [28–30], ammonium thiocyanate [31–33], etc. Because the band gap of g-C₃N₄ is 2.7 eV, it can absorb visible light shorter than 450 nm effectively, implying broad prospects in solar energy conversion applications. Due to the aromatic C-N heterocycles, g-C₃N₄ is thermally stable up to 600 °C in air. Moreover, g-C₃N₄ is insoluble in acids, bases or organic solvents, exhibiting good chemical stability.

However, some bottlenecks in the photocatalytic activity of g-C₃N₄ still exist, such as fast photogenerated carrier recombination, limited active site, small specific surface area, low light absorption capacity, unsatisfactory crystallinity and unignorable surface

defects. How to promote the efficient migration and separation of photogenerated carriers, expand the spectral response range and increase the specific surface area of g-C$_3$N$_4$ is the core problem to achieve high energy conversion efficiency. In practice, the introduction of impurities into the g-C$_3$N$_4$ matrix through copolymerization and doping has become an effective strategy to change the electronic structure and band structure of g-C$_3$N$_4$. On the other hand, numerous research works have demonstrated that the physicochemical properties and photocatalytic efficiency of the polymer g-C$_3$N$_4$ can be significantly improved by optimizing synthesis techniques such as supramolecular and copolymerization techniques with identical structural and nano-structural designs, or by template-assisted methods to improve porosity and surface area [34–39]. Amongst various modification approaches, designing and constructing a more suitable band structure is the most important prerequisite to improve the charge separation efficiency, thereby enhancing the photocatalytic performance.

In this review, the synthesis parameters of g-C$_3$N$_4$ are discussed first, mainly including the types of g-C$_3$N$_4$ precursors, reaction temperature, reaction atmosphere and reaction duration. The influence of different synthesis parameters of g-C$_3$N$_4$ are compared and summarized in detail here. Then, the construction approaches of various nanostructures are reviewed, such as hard and soft template, supramolecular preorganization, template-free approaches, etc. It again manifests that the specific surface area and photocatalytic efficiency of g-C$_3$N$_4$ can be directly manipulated by means of different nanostructure design approaches. Furthermore, the characteristics of different exfoliation methods are summarized for stark comparisons. Liquid exfoliation of bulk g-C$_3$N$_4$ has gradually become the most popular exfoliation method. The overall framework of synthesis and properties of g-C$_3$N$_4$ for enhanced photocatalytic performance are illustrated in Figure 1. Finally, the problems of g-C$_3$N$_4$ materials in photocatalysis and the prospect of further development are proposed, which may be favorable to the design of more efficient and practical g-C$_3$N$_4$ or g-C$_3$N$_4$-based photocatalyst.

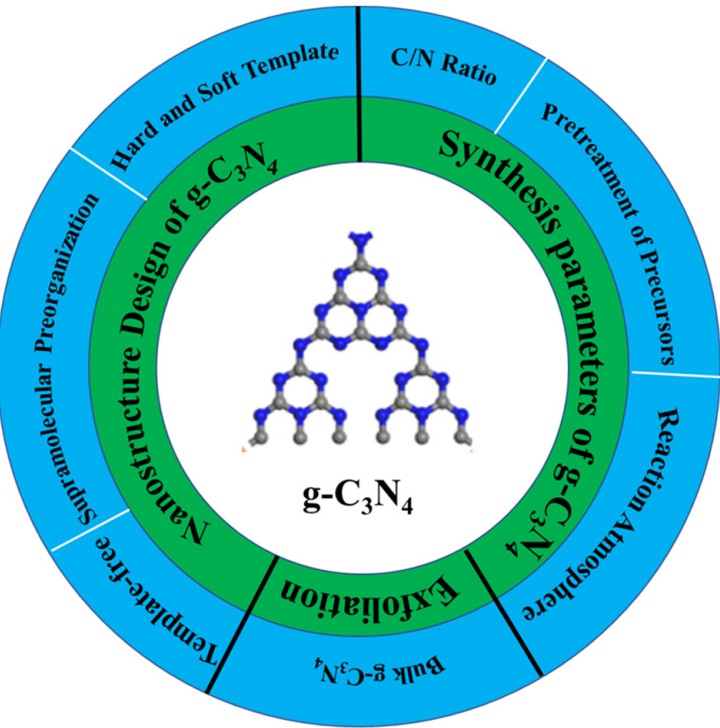

**Figure 1.** Overall framework of synthesis and properties of g-C$_3$N$_4$ for enhanced photocatalytic performance.

## 2. Influence of Synthesis Parameters

### 2.1. Precursors and Reaction Temperature

The first reported g-C₃N₄ as a heterogeneous catalysis was present in 2006 [40]. Subsequently, various precursors such as urea [41], thiourea [42], cyanamide [43,44] and dicyandiamide [45,46] have been employed to synthesize g-C₃N₄ by thermal treatment methods. In 2009, Wang et al. firstly used cyanamide as the precursor of g-C₃N₄ for producing hydrogen from water under visible-light irradiation in the presence of a sacrificial donor [47]. This pioneering work represents an important first step towards photosynthesis in general, where artificial conjugated polymer semiconductors can be used as energy transducers. In order to demonstrate the reaction intermediate compounds, characterization techniques such as thermogravimetric analysis (TGA) and X-ray diffraction (XRD) are used to characterize the reaction. Figure 2a displayed that the graphitic planes are constructed from tri-s-triazine units connected by planar amino groups. Figure 2b is the XRD pattern of the obtained g-C₃N₄ powder. From the ultraviolet-visible spectrum (Figure 2c), it can be seen that the band gap of g-C₃N₄ is 2.7 eV. The synthesis of g-C₃N₄ was a combination of polyaddition and polycondensation. At a reaction temperature of 203 and 234 °C, the cyanamide molecules can be condensed to dicyandiamide and melamine, respectively. The ammonia is then removed by condensation. When the temperature reaches 335 °C, large amounts of melamine products are detected. When further heating to 390 °C, the rearrangements of melamine will result in the formation of tri-s-triazine units. Finally, when heating to 520 °C, the polymeric g-C₃N₄ are synthesized via the further condensation of the unit. However, g-C₃N₄ will be unstable at above 600 °C. Furthermore, when the reaction degree is higher than 700 °C, g-C₃N₄ will decompose. Figure 2d shows the structural phase transition process from cyanamide to g-C₃N₄ at different temperatures. In addition to in-situ characterization experiments to verify the reaction process, it can also be demonstrated by relevant simulation calculations. The first-principles DFT calculations were performed using a plane wave basis set with a 550 eV energy cutoff [40]. The calculation results showed that the cohesion energy increased under the addition of multiple reaction pathways, which confirmed that melamine was produced upon heating the cyanamide, as shown in Figure 2e. In another work, Ang et al. demonstrated that when thiourea is used as the precursor and TiO₂ or SiO₂ are used as the inorganic substrate, the melon nanocomposites can be formed at a low temperature of 400 °C [48]. However, the degree of polymerization of g-C₃N₄ in the nanocomposites is low, so its photocatalytic performance is moderate. Following this, Zhang et al. reported a simple method of g-C₃N₄ synthesis from thiourea without the aid of any substrates [49]. It was found that the obtained g-C₃N₄ exhibited a good condensation likely due to the presence of sulfur species in thiourea, thus accelerating the degree of polymerization and condensation of thiourea at high temperatures. Having insights into the results of different reports, it can be found that different precursors have distinct characteristics and advantages. In order to obtain well-condensed g-C₃N₄ with high quality, the most crucial point is to select the best reaction temperature corresponding to the selected precursor.

In order to prove that g-C₃N₄ has been successfully prepared, a variety of analytical measurements can be used, such as X-ray photoelectron spectroscopy (XPS), XRD and Fourier transform infrared (FTIR) spectroscopy. The basic experimental procedures for the synthesis, characteristics of different precursors and characterization of g-C₃N₄ are described above. What effects other synthesis parameters have on g-C₃N₄ will be compared and discussed in detail in the following sections.

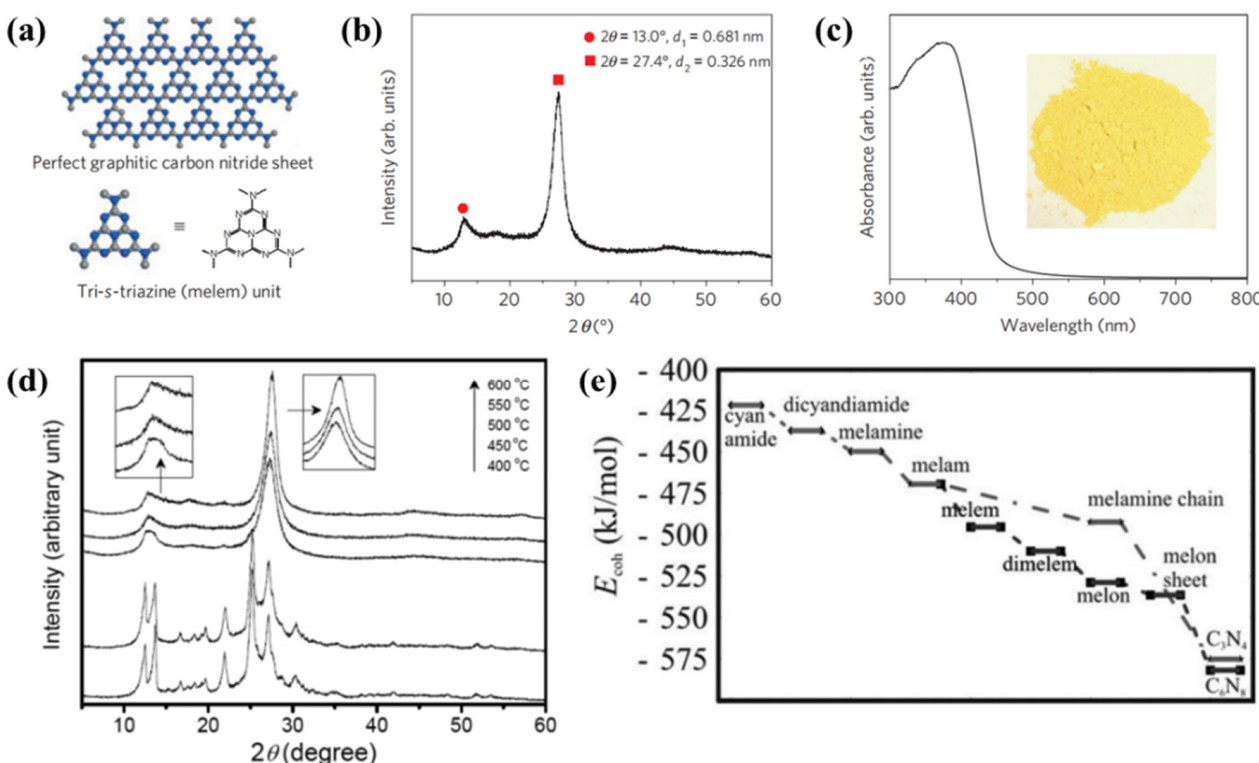

**Figure 2.** Crystal structure and optical properties of g-$C_3N_4$. (**a**) Schematic diagram of a perfect g-$C_3N_4$ sheet constructed from melem units. (**b**) Experimental XRD pattern of polymeric carbon nitride, revealing a graphitic structure with an interplanar stacking distance of the aromatic unit (0.326 nm). (**c**) Diffuse reflectance spectrum of the polymeric carbon nitride. Inset: Photograph of the photocatalyst. (**d**) XRD patterns of g-$C_3N_4$ treated at different temperatures. (**e**) Calculated energy diagram for the development of g-$C_3N_4$ using cyanamide precursor [47].

*2.2. C/N Ratio*

Generally, g-$C_3N_4$ exhibits a high physicochemical stability and ideal band structure, due to the high condensation degree and the presence of the heptazine ring structure. When the appropriate precursor and condensation method are selected, the C/N ratio in layered g-$C_3N_4$ is about 0.75. Many studies have confirmed that when the precursors and synthesis parameters of g-$C_3N_4$ are changed, the physicochemical properties of g-$C_3N_4$ will be significantly affected, such as band gap width, specific surface area, C/N ratio, etc., which will directly affect the photocatalytic efficiency and other applications' performance [50–54]. Yan et al. synthesized g-$C_3N_4$ by directly heating the low-cost melamine, and they change the C/N ratio by controlling different heating temperatures [55]. The research showed that when the heating temperatures increased from 500 to 580 °C, the ratio of C/N increased from 0.721 to 0.742. Meanwhile, the band gaps of g-$C_3N_4$ decreased from 2.8 to 2.75 eV. Apparently, increasing the heat temperature will decrease the photooxidation ability of g-$C_3N_4$. The C/N ratio, i.e., the degree of condensation, is inconsistent with the structural integrality. Therefore, in order to obtain a better photocatalytic efficiency, reasonable optimization measures must be taken to make the C/N ratio closer to the ideal ratio (0.75) [56]. Therefore, the presence of trace amino groups is actually conducive to improving the surface activity of g-$C_3N_4$, thus exhibiting better interaction with the reactant molecules [8,57,58]. However, due to incomplete concentration, the C/N ratio is lower than the ideal value (0.75), resulting in a large number of defects, inhibiting the rate of carrier transport and separation, and seriously reducing the photocatalytic efficiency. Therefore, in order to obtain a relatively high photocatalytic efficiency, the optimal C/N ratio is 0.75, which can usually reflect that g-$C_3N_4$ has a good concentration and stability.

## 2.3. Pretreatment of Precursors

It has been shown that modification and pretreatment of nitrogen-rich precursors before thermal annealing can effectively improve the physicochemical properties of g-C$_3$N$_4$. One of the effective pretreatment methods is by acid treatment. Yan et al. reported the synthesis of g-C$_3$N$_4$ by directly heating the sulfuric-acid-treated melamine precursor [59]. It is worth noting that the carbon nitride synthesized from sulfuric acid treated melamine (15.6 m$^2$/g) shows relatively higher BET surface area than that of samples synthesized from untreated melamine (8.6 m$^2$/g). The reason can be attributed to the effect of pretreatment of melamine with H$_2$SO$_4$ on its condensation process, during which sublimation of melamine is inhibited significantly. In addition to the pretreatment of melamine with H$_2$SO$_4$, HCl and HNO$_3$ also exhibited good pretreatment effects on melamine [60–63].

In addition to acid precursors, pretreatment methods of sulfur-mediated synthesis can also be used to regulate the structure and physicochemical properties of g-C$_3$N$_4$ [64]. The fundamental reason is that the presence of the sulfur group in the sulfur-containing thiourea provides an additional chemical pathway to regulate the degree of condensation and polymerization of g-C$_3$N$_4$ because it is easy to leave the -SH groups. Zhang et al. advanced this strategy by employing cheap and easily available elemental sulfur as the external sulfur species instead of sulfur-containing precursors for the sulfur-mediated synthesis of g-C$_3$N$_4$ photocatalysts [65]. In comparison with unmodified g-C$_3$N$_4$, the vibrations of g-C$_3$N$_4$-S$_x$ are less intensive when increasing the amount of elemental sulfur (S$_8$), especially for the broad band at 2900–3300 cm$^{-1}$. Thus, S$_8$-mediated synthesis helps to advance the polymerization of melamine precursors, leaving fewer amino-containing groups as surface defects. In this report, the structure, electronic and optical properties of g-C$_3$N$_4$ have been effectively modified, and its physicochemical properties have also been significantly improved. Under visible light irradiation at 420 nm, the photocatalytic activity of water reduction and oxidation was enhanced.

## 2.4. Reaction Atmosphere

In addition to the types of precursors, reaction temperature and duration, the physico-chemical properties and structure of g-C$_3$N$_4$ are also strongly influenced by the reaction atmosphere, because the reaction atmosphere can induce a variety of defects and carbon and nitrogen vacancies. In fact, defects are essential for catalytic reactions because they can act as active sites for reactant molecules and change the band structure by introducing additional energy levels in the forbidden band, thus extending the spectral absorption range [66–69]. By controlling the polycondensation temperature of a dicyandiamide precursor in the preparation of g-C$_3$N$_4$, Niu et al. introduced nitrogen vacancies in the framework of g-C$_3$N$_4$ [70]. The excess electrons caused by nitrogen loss in g-C$_3$N$_4$ lead to a large number of C$^{3+}$ states associated with nitrogen vacancies in the band gap, thus reducing the intrinsic band gap from 2.74 eV to 2.66 eV. Steady and time-resolved fluorescence emission spectra show that, due to the existence of abundant nitrogen vacancies, the intrinsic radiative recombination of electrons and holes in g-C$_3$N$_4$ is greatly restrained, and the population of short-lived and long-lived charge carriers is decreased and increased, respectively. In another study, Niu et al. produced a novel visible light photocatalyst R-melon by heating the melon in a hydrogen atmosphere [71]. Compared to the pristine melon with a bandgap of 2.78 eV, the resultant R-melon with a bandgap of 2.03 eV has a widened visible light absorption range and suppressed radiative recombination of photo-excited charge carriers due to homogeneous self-modification with nitrogen vacancies. Table 1 shows the summary of the specific surface areas and band gaps of g-C$_3$N$_4$ photocatalysts developed from various precursors and different reaction parameters. The results demonstrated that the band structure, electronic properties and specific surface area of g-C$_3$N$_4$ can be changed by adjusting the reaction parameters and precursors, so as to improve its photocatalytic performance.

**Table 1.** Precursors and Reaction Parameters Employed in the g-C$_3$N$_4$ Synthesis.

| Precursors | Synthesis Parameters | Surface Area (m$^2$ g$^{-1}$) | Band Gap Energy (eV) | Ref. |
|---|---|---|---|---|
| Cyanamide | 550 °C, 4 h, air | 10 | 2.7 | [47] |
| Dicyandiamide | 550 °C, 3 h, air | 12.3 | 2.66 | [56] |
| Dicyandiamide | 600 °C, 4 h, air | 12.8 | 2.75 | [72] |
| Dicyandiamide | 550 °C, 4 h, H$_2$ | 20.91 | 2.0 | [73] |
| Melamine | 500 °C, 2 h, air | 7.1 | 2.83 | [74] |
| Melamine | 520 °C, 2 h, N$_2$ | 17.4 | 2.74 | [75] |
| H$_2$SO$_4$-treated melamine | 600 °C, 4 h, Ar | 15.6 | 2.69 | [59] |
| HCl-treated melamine | 550 °C, 4 h, air | 26.2 | 2.73 | [61] |
| Thiourea | 550 °C, 3 h, air | 11.3 | 2.6 | [56] |
| Urea | 450 °C, 2 h, air | 135.6 | 2.76 | [76] |

## 3. Morphology and Structure Design of g-C$_3$N$_4$

### 3.1. Hard and Soft Template Approach

Apart from regulating the synthesis parameters, introducing nano-templates and nano-casting with different morphology and ordered porosity on the basis of bulk g-C$_3$N$_4$ is another promising method to change the morphology and structural characteristics of g-C$_3$N$_4$ structure and the interlayer interaction. As a matter of fact, researchers have effectively designed controllable nanostructures for g-C$_3$N$_4$ through hard template or soft template methods, such as porous g-C$_3$N$_4$, one-dimensional nanostructures, hollow g-C$_3$N$_4$ nanospheres, etc [11,77–85]. It has been proven that the porosity, structure, morphology, surface area and size can be easily controlled by adjusting the appropriate template. Moreover, the larger surface area and more active sites are generally more favorable for photocatalytic applications of g-C$_3$N$_4$.

The hard template method is almost identical to the traditional casting process and is one of the most common techniques for developing nanostructured g-C$_3$N$_4$ materials. In this way, the various structures and geometries of g-C$_3$N$_4$ can be designed using hard templates as needed, and their length scales are usually around nanometers and microns. The most typical structure-oriented agent is a silica template with a controllable nanostructure. The early study on the mesoporous g-C$_3$N$_4$ synthesized using cyanamide as a precursor and silica nanoparticles with a size of 12 nm as a template was reported by Goettmann et al. [40]. The results show that the silica nanoparticles can be uniformly dispersed in the cyanamide monomer, which is due to the appropriate surface interaction between the silica surface and the amine and aromatic nitrogen groups. After heating treatment and cyanamide condensation, the g-C$_3$N$_4$/silica hybrid is formed, and well-dispersed silica nanoparticles are preserved in the g-C$_3$N$_4$ matrix. Ammonium hydrogen fluoride (NH$_4$HF$_2$) solution can be used to remove the silicon template. The average diameter is 12 nm, and the surface area in the range of 86 to 439 m$^2$ g$^{-1}$ can be regulated by adjusting the mass ratios of silica/cyanamide from 0 to 1.6. In another work carried out by Yuki Fukasawa et al., uniform-sized silica nanospheres (SNSs) assembled into close-packed structures were used as a primary template for ordered porous g-C$_3$N$_4$, which was subsequently used as a hard template to generate regularly arranged Ta$_3$N$_5$ nanoparticles of well-controlled size [86]. The cyanamide is infiltrated and polymerized in the narrow void of SNSs to form porous g-C$_3$N$_4$, and then the SNSs is removed by HF treatment, as shown in Figure 3a. Therefore, the resulting g-C$_3$N$_4$ has an anti-opal structure, and the size of the spherical hole indicates the size of the SNSs used, as shown in the SEM images of Figure 3b–e. In this study, the pore size of g-C$_3$N$_4$ was between 50 and 80 nm. In spite of the silica hard template, Chen et al. reported the synthesis of porous g-C$_3$N$_4$ by using multi-walled carbon nanotube (CNT) as a novel hard template [87]. Unlike other hard templates, CNT can be easily removed and recovered by ultrasonic methods, resulting in a relatively simple preparation of porous g-C$_3$N$_4$.

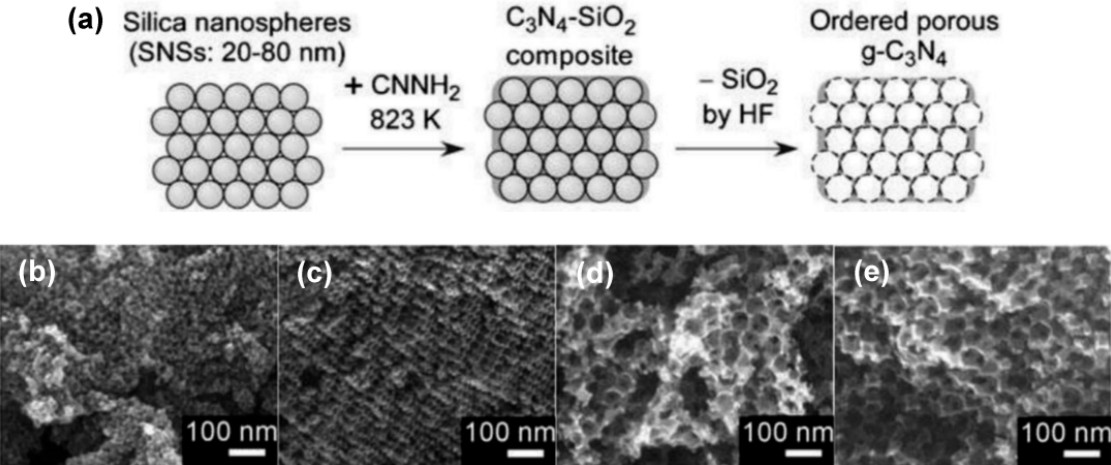

**Figure 3.** (**a**) Synthesis procedure of ordered porous g-C$_3$N$_4$. Field emission SEM (FESEM) images of porous g-C$_3$N$_4$ prepared using silica spheres with various diameters: (**b**) 20, (**c**) 30, (**d**) 50, and (**e**) 80 nm [86].

It can be seen that during the synthesis of g-C$_3$N$_4$ through hard templates, extremely dangerous, toxic and expensive fluorine-containing etchers (such as HF and NH$_4$HF$_2$) are used to remove the sacrificial templates. This greatly limits the practical application of the method in large-scale industrial processes. Therefore, apart from the hard template synthesis method of g-C$_3$N$_4$, the relatively "environmentally friendly" soft template process can not only change the morphology and structure of g-C$_3$N$_4$ through the selection of multiple soft templates, but also simplify the synthesis route of g-C$_3$N$_4$ [88,89]. Different from the hard template method, in the soft template method, the nano-structure g-C$_3$N$_4$ is synthesized by soft structure guiding agents such as ionic liquid, amphiphilic block polymer and surfactant, so as to rationally design g-C$_3$N$_4$ with a highly porous nanostructure [90]. For instance, bimodal mesoporous g-C$_3$N$_4$ is synthesized using Triton X-100 as a soft-template and melamine and glutaraldehyde as precursors through polymerization and carbonization [91]. The results show that the mesopore sizes in the g-C$_3$N$_4$ are centered at 3.8 nm and 10–40 nm. The former was attributed to the removal of the Triton X-100, while the latter was ascribed to the aggregates of plate-like g-C$_3$N$_4$. In another study reported by Wang et al., a variety of soft-templates (e.g., Triton X-100, P123, F127, Brij30, Brij58, and Brij76) as well as some ionic surfactants are tested as structure-directing agents for the synthesis of mesoporous g-C$_3$N$_4$ [90]. Most as-prepared g-C$_3$N$_4$ materials possess a high surface area. Moreover, this work points out two problems. The first was that only a few selected soft templates (Triton X-100 and ionic liquids) led to the presence of mesoporous g-C$_3$N$_4$ structures with high surface areas. Due to the premature decomposition of the template material, the pores of g-C$_3$N$_4$ are easily resealed. The second problem is that the obtained g-C$_3$N$_4$ contains a large amount of carbon elements from the template polymer, which significantly changes the morphology and structure of g-C$_3$N$_4$, thereby reducing its photocatalytic activity.

Overall, it is not difficult to find that the synthesis of g-C$_3$N$_4$ with various nanostructures assisted by hard and soft templates is a facile and efficient approach. Nowadays, researchers are still developing a variety of new templates to achieve more interesting g-C$_3$N$_4$ nanostructures with high photocatalytic efficiency.

*3.2. Supramolecular Preorganization Approach*

In contrast to the previously discussed hard and soft template synthesis approaches, molecular self-assembly is a self-templating approach (namely supramolecular preorganization approach) in which molecules spontaneously form a stable g-C$_3$N$_4$ structure from non-covalent bonds under equilibrium conditions in the absence of an external template [80,92–94]. Recently, supramolecular preassembly of triazine molecules has become an

interesting method to regulate the structural, textural, optical, and electronic features of g-$C_3N_4$, thus affecting its photocatalytic activity [95–98]. For example, nanostructured g-$C_3N_4$ materials can be developed by supramolecular preorganization of melamine precursors to triazine derivatives to form hydrogen bond molecular assemblies, i.e., melamine–cyanuric acid, melamine–trithiocyanuric acid mixtures or their derivatives [99–101]. Jun et al. first synthesized g-$C_3N_4$ by molecular cooperative assembly between triazine molecules [102]. Flower-like, layered spherical aggregates of melamine cyanuric acid complex (MCA) are formed by precipitation from equimolecular mixtures in dimethyl sulfoxide (DMSO). The oxygen-containing intermolecular structure connected by hydrogen bonding and stacked in graphitic fashion facilitates the condensation process and enables structural perfection. The obtained material synthesized by this supramolecular preorganization approach has stronger spectral absorption (an increased band gap of 0.16 eV), and the photogenerated carrier lifetime is extended nearly twice as long as that of bulk g-$C_3N_4$.

It can be seen that the combination of two or more monomers in different solvents can form supramolecular complexes. These supramolecular complexes are usually linked by hydrogen bonds. Therefore, it is expected that the addition of new monomers to hydrogen-bonded supramolecular complexes as "terminators" will be an attractive technique to further adjust the morphology, photophysical properties, and electronic band structure of g-$C_3N_4$.

### 3.3. Template-Free Approach

Compared with the hard and soft template synthesis approaches, the template-free approach has unique advantages, such as no need for various high-cost and dangerous templates containing fluorine, and no residue of any template components. Indeed, many studies have proven that g-$C_3N_4$ nanostructure designs with a variety of morphologies and desired sizes, such as nanorods, quantum dots, microspheres, nanofibers, etc., can also be achieved using a template-free approach. Bai et al. reported that the transformation of g-$C_3N_4$ from nanoplates to nanorods was realized by a simple reflux method [103]. Various aspect ratios were achieved by changing the reflux duration and solvent ratio without the assistance of any templates. As per the TEM images shown in Figure 4a–d, most irregular nanoplates are observed in the untreated g-$C_3N_4$ sample, while the refluxed g-$C_3N_4$ sample mainly contains nanorods. The length of g-$C_3N_4$ nanorods is 0.5–3 μm. The growth mechanism and process can be summarized as follows (Figure 4e): under ultrasonic action, $CH_3OH$ can strip g-$C_3N_4$ bulks into nanosheets. The reflux process can remove surface defects and transform them into nanorods. After reflux treatment, the g-$C_3N_4$ nanosheets showed the metastable state. With the increase of time, the layered structure of g-$C_3N_4$ will curl and finally grow into nanorods. Finally, the shorter g-$C_3N_4$ nanorods are redissolved into the solution, while the longer nanorods will continue to grow during reflux treatment. The increase in visible light degradation activity of methylene blue can be induced for the increase of active lattice face and elimination of surface defects.

Additionally, Wang et al. described a facile and generally feasible method to synthesize nanotube-type g-$C_3N_4$ by directly heating melamine packed in an appropriate compact degree without templates [104]. A certain amount of melamine was placed into a semi-closed alumina crucible followed by consecutively shaking the crucible using a vibrator at a fast rate to achieve a moderately compact packing degree. This process is very crucial for the synthesis of nanotube-type g-$C_3N_4$. TEM images show that the wall thickness of the nanotubes in the bulk phase is about $15 \pm 2$ nm, while the inner diameter is about $18 \pm 2$ nm. In this report, the formation process and mechanism of nanotubes have been studied in detail. During the pyrolysis process, melamine releases $NH_3$ gas, which passes through the stacked melamine layers to form rolled g-$C_3N_4$ nanosheets. Due to the need to reduce the surface free energy, the nanosheets eventually bend into the form of nanotubes. It is worth pointing out that when the melamine layer is loosely stacked, g-$C_3N_4$ cannot form a nanotube structure, which indicates that obtaining the appropriate packing degree of g-$C_3N_4$ is crucial for the formation of the nanotube structure.

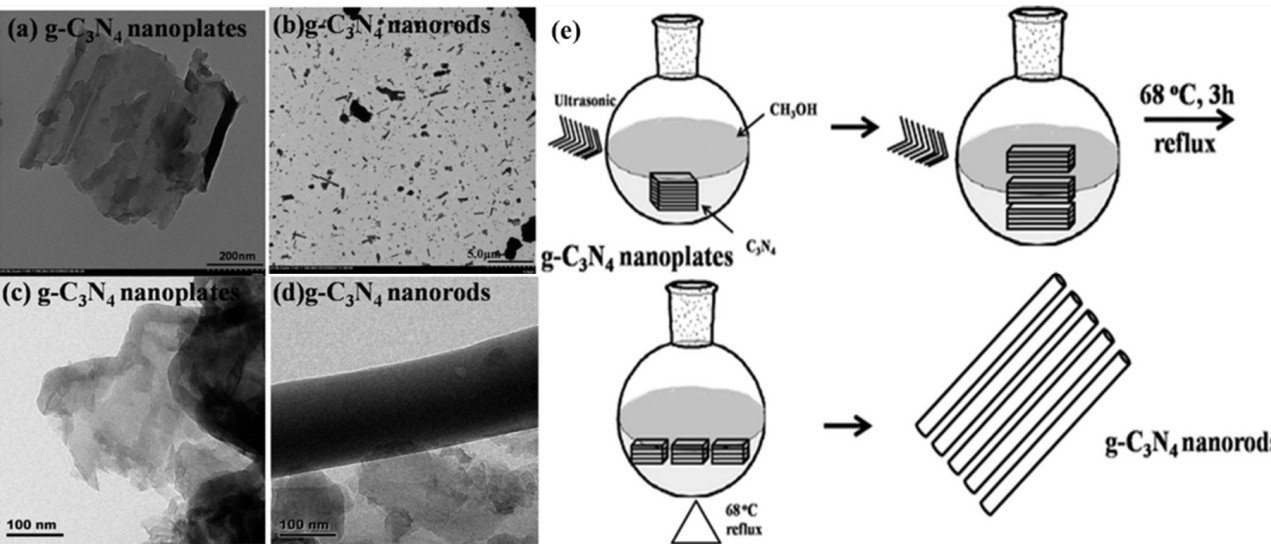

**Figure 4.** TEM images of g-$C_3N_4$ nanoplates (**a**,**c**) and nanorods (**b**,**d**). (**e**) Schematic illustration of the formation processes of g-$C_3N_4$ nanorods from g-$C_3N_4$ nanoplates [103].

## 4. Exfoliation of Bulk g-$C_3N_4$

Although the specific surface area of the monolayer g-$C_3N_4$ is theoretically large, the specific surface area of the block is very low indeed, usually less than 10 m$^2$ g$^{-1}$, due to the stacking of g-$C_3N_4$ layers [105]. Therefore, delaminating g-$C_3N_4$ into several layers is a promising way to improve photocatalytic performance and produce more interesting surface, optical and electronic properties [106–109]. There are many methods of exfoliating g-$C_3N_4$, such as ultrasonication-assisted liquid exfoliation, the liquid ammonia-assisted lithiation and the post-thermal oxidation etching route [17,108–115]. Similar to most two-dimensional materials, there is a weak van der Waals force between the layers of g-$C_3N_4$. Thus, the van der Waals force can be effectively overcome and the separation between layers can be achieved by means of energy assistance such as ultrasound in an appropriate solvent.

Liquid exfoliation is simple and convenient, and has gradually become the most commonly used exfoliation method by most researchers. Yang et al. demonstrated the synthesis of free-standing g-$C_3N_4$ nanosheets by liquid phase exfoliation [110]. The method uses g-$C_3N_4$ powder as a starting material and various organic solvents (such as isopropanol (IPA), N-methyl-pyrrolidone (NMP), acetone, and ethanol) as dispersing media. As shown in Figure 5a,b, many nanosheets with laminar morphology like silk veil can be observed, which is in stark contrast from that of bulk g-$C_3N_4$. The photographs in Figure 5e clearly demonstrated that NMP is a promising dispersing solvent among various dispersing solvents. It can stably disperse individual nanosheets; however, the disadvantage is that the boiling point of NMP is relatively high, which is difficult to remove. The agglomeration phenomenon of g-$C_3N_4$ layer can be observed in NMP. In comparison, low-boiling-point IPA is the best medium for preparing g-$C_3N_4$. The obtained g-$C_3N_4$ in IPA exhibited a surface area of up to 384 m$^2$ g$^{-1}$ and a thickness of about 2 nm, as shown in Figure 5c,d. Moreover, the stability of IPA is superior, and there is no precipitation phenomenon after 4 months of storage (Figure 5f).

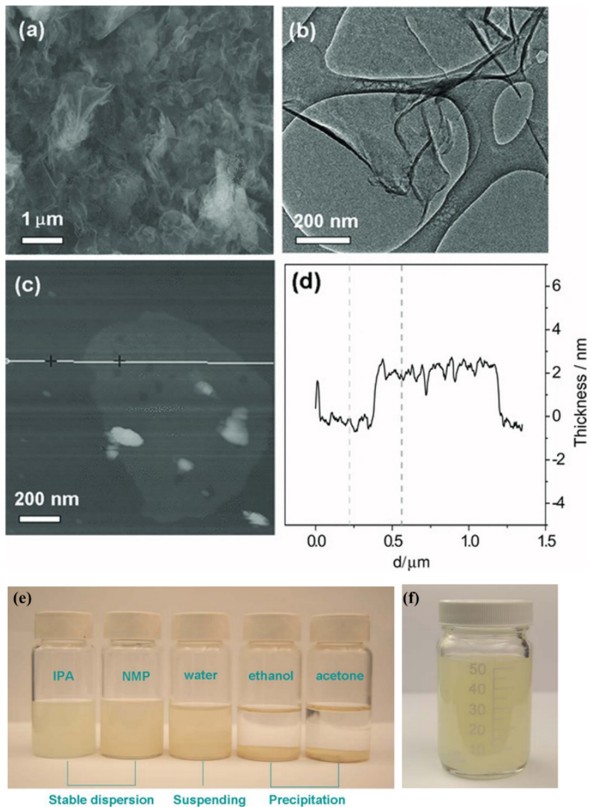

**Figure 5.** Typical FESEM (**a**) and TEM (**b**) images unveiling the flexible g-C$_3$N$_4$ nanosheets with the size from 500 nm to several micrometers. Representative AFM image (**c**) and corresponding thickness analysis (**d**) taken around the white line (**c**) revealing a uniform thickness of about 2 nm for g-C$_3$N$_4$ nanosheets. (**e**) Photographs of the g-C$_3$N$_4$ nanosheet dispersions in IPA, NMP, water, ethanol and acetone, respectively, after 2 days storage under ambient conditions, and (**f**) in IPA solvent after 4 months storage [110].

## 5. Doping of g-C$_3$N$_4$

It is well-known that g-C$_3$N$_4$ is a metal-free n-type semiconductor. Due to the high ionization energy and high electronegativity of metal-free semiconductors, it is easy for them to form covalent bonds with other compounds by obtaining electrons during the reaction. In order to maintain this unique advantage of metal-free semiconductors, researchers have implemented a series of non-metal doping g-C$_3$N$_4$, including oxygen, phosphorus, sulfur, carbon, halogen, nitrogen and boron [116–121]. For instance, O-doping is a facile method to improve the photocatalytic ability of g-C$_3$N$_4$. Zeng et al. synthesized one-dimensional porous architectural g-C$_3$N$_4$ nanorods by direct calcination of hydrous melamine nanofibers precipitated from an aqueous solution of melamine [122]. The porous structure increases the specific surface area, enhances the light absorption capacity and improves the catalytic reaction rate. At the same time, doping oxygen atoms into the g-C$_3$N$_4$ matrix breaks the symmetry of the pristine structure, making more efficient separation of electron/hole pairs. In general, non-metallic doping usually changes the surface morphology and structure of g-C$_3$N$_4$, thereby affecting the light absorption efficiency and regulating the catalytic efficiency.

Different from non-metal doping, metal-doped g-C$_3$N$_4$ has the advantages of reducing band gap to enhance visible light absorption and to improve catalytic performance. Commonly used doped metals include alkali metals (Li, Na, K) and transition metals (Fe, Cu, and W) [123–127]. For example, Xiong et al. reported that the doping of alkali metals (K, Na) in the g-C$_3$N$_4$ framework significantly increases the transfer and separation rates of photogenerated carriers and induces more efficient redox catalytic reactions [128]. DFT calculation results showed that K or Na doping can reduce the band gap energy of

g-$C_3N_4$. Generally, the introduction of metal ions can form new energy levels, increase the specific surface area, and sometimes resist the recombination of charge carriers produced by photons. The catalytic efficiency of noble metal doping is usually higher, but the photodegradation efficiency of some ordinary transition-metal-doped g-$C_3N_4$ for some pollutants can be comparable to that of noble-metal-doped g-$C_3N_4$ when the appropriate doping amount and mode are selected.

## 6. Applications of g-$C_3N_4$

Due to its moderate energy gap, excellent electronic properties, rich functional groups and surface defects, g-$C_3N_4$ can be widely used in environmental treatment and pollutant degradation, including water splitting, hydrogen generation, $CO_2$ conversion and organic pollutants degradation [129–132].

As reported, the pristine g-$C_3N_4$ has limitations such as small specific surface area and fast charge recombination rate, which leads to a low water splitting ability of g-$C_3N_4$. To solve this problem, Chen et al. improved the water splitting capacity via adjusting the dimension of g-$C_3N_4$ [133]. As shown in Figure 6a,b, they demonstrated that the evolution rates of $H_2$ and $O_2$ of three-dimensional porous g-$C_3N_4$ in visible light are significantly higher than those of the pristine g-$C_3N_4$, reaching 101.4 and 49.1 µmol $g^{-1}$ $h^{-1}$, respectively. Fu et al. reported oxygen-doped g-$C_3N_4$ [134]. As exhibited in Figure 6c-d, the O-doped g-$C_3N_4$ has a narrower band gap and greater $CO_2$ affinity, which significantly improves the photogenerated carrier separation efficiency and $CO_2$ conversion ability. In addition, a lot of efforts have been made to enhance its photocatalytic activity to improve its pollutant degradation ability. As shown in Figure 6e-f, Dou et al. reported that mesoporous g-$C_3N_4$ has a strong ability to remove antibiotics under visible light [135], which is mainly due to the porous structure that improved the utilization of light.

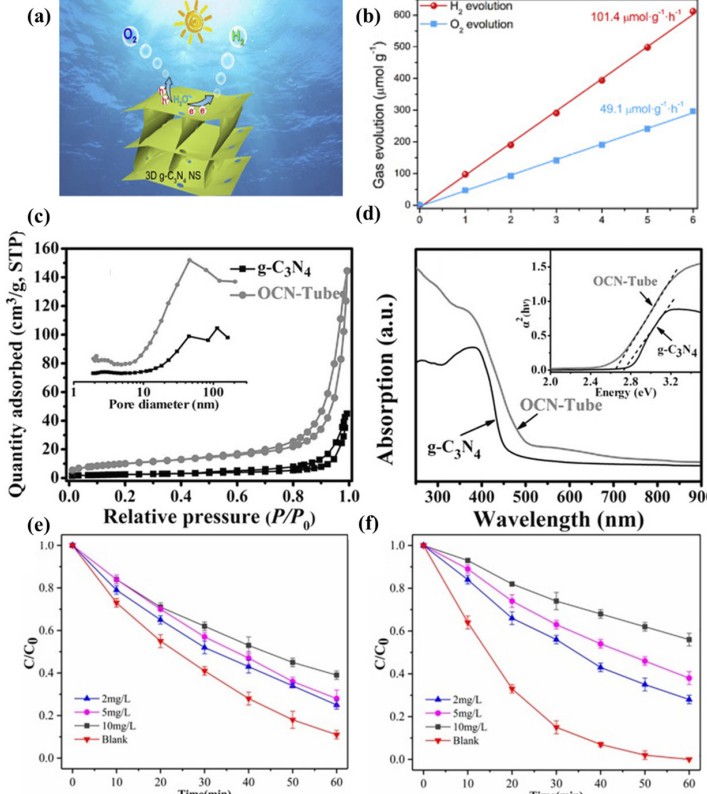

**Figure 6.** (**a**) Water splitting for $H_2$ and $O_2$ Evolution. (**b**) Time−dependent overall water splitting over 3D g-$C_3N_4$ [133]. (**c**) $N_2$ adsorption–desorption isotherms and corresponding pore size distribution curves. (**d**) UV–vis diffuse reflectance spectra [134]. The effects of humic acid on (**e**) amoxicillin and (**f**) cefotaxime photodegradation by mesoporous carbon nitride (initial pH = 7) [135].

### 7. Conclusions and Prospects

In summary, due to the merits of low cost, high stability and visible light response, g-$C_3N_4$ is one of the most promising photocatalytic materials to replace $TiO_2$. This review mainly introduces the synthesis of g-$C_3N_4$, the improvement of the g-$C_3N_4$ crystal structure, the light absorption enhancement, structure design optimization, as well as the improvement of electronic properties and optimization of energy band, so as to promote the photocatalytic application of g-$C_3N_4$. As summarized above, the appropriate reaction temperature and duration of the condensation process are beneficial to improve the crystallinity of g-$C_3N_4$. Various desirable nanostructures of g-$C_3N_4$ can be constructed via hard and soft template approaches, supramolecular preorganization approach, and template-free approach. Liquid exfoliation of bulk g-$C_3N_4$ has becoming the most facile and promising method to improve the surface area of g-$C_3N_4$.

Therefore, in order to synthesize the ideal g-$C_3N_4$ with high photocatalytic efficiency, it is necessary to pay attention to the following crucial elements: (i) Controlling the corresponding reaction temperature and reaction time according to the selected precursor material; (ii) Controlling the C/N ratio close to 0.75 and the band gap to 2.7 eV; (iii) Extending the specific surface area by selecting suitable nanostructure design approaches.

It can be assumed that in the future, g-$C_3N_4$, a booming photocatalytic hot spot material, will face unlimited opportunities and challenges. Although g-$C_3N_4$ can be easily synthesized by thermal polymerization of nitrogen-rich precursors, its photocatalytic efficiency is not high due to its small specific surface area, limited surface reaction sites, and insufficient utilization of broad-spectrum sunlight. Here, some pivotal issues are elaborated in the following:

(i) So far, the use of visible and near-infrared light is far from sufficient. On the one hand, it depends on the synthesis and structural design of the g-$C_3N_4$ material; on the other hand, a third or more component with a suitable band structure can also be cleverly added to design the interface electronic structure and expand the optical absorption region.

(ii) Due to the limited surface area of g-$C_3N_4$ and the inevitable defects on the surface, the smooth transfer of photogenerated carriers is hindered. Therefore, the synthesis approached needs to be further developed. In order to optimize photocatalytic performance, some environmentally friendly, simple and efficient synthesis routes are urgently required.

(iii) The integration of photocatalytic applications in many fields into one photocatalytic system is a widely used catalytic process. This requires the hybridization of multifunctional materials with reasonable energy structure. How to optimize the energy band structure and surface topography of g-$C_3N_4$ reasonably to improve the compatibility of energy conversion and environmental protection is very promising.

(iv) More theoretical studies about g-$C_3N_4$ need to be combined with practical catalytic applications. It is certain that in-depth fundamental theory based on physical chemistry research, in collaboration with laboratory findings, will positively promote the advances in materials science and technology.

**Author Contributions:** Conceptualization, S.Z. and D.Z.; investigation, H.L.; writing—original draft preparation, J.P. and D.Y. All authors have read and agreed to the published version of the manuscript.

**Funding:** This research received no external funding.

**Conflicts of Interest:** The authors declare no conflict of interest.

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
