# Peer review of "Recent Advances in g-C3N4 Photocatalysts: A Review of Reaction Parameters, Structure Design and Exfoliation Methods"

_catalysts, doi:10.3390/catal13111402_

Round 1

Reviewer 1 Report

Comments and Suggestions for Authors

Dear Authors,

I have reviewed your Review manuscript “Recent Advances in g-C3N4 Photocatalysts: A Review of Reaction Parameters, Structure Design and Exfoliation Methods” and expressed my positive feedback regarding your submission. You have explored a very interesting and important topic, but it should be pointed out more, so I request major revisions.

I have a few suggestions for revisions that I believe would further enhance the quality and comprehensiveness of the manuscript.

 ·        Addition of "Doping Strategies" Section:

I recommend including a dedicated section that discusses the various doping strategies used in g-C3N4 photocatalysis. This is an essential aspect of the research and will give readers a deeper understanding of the topic.

 ·        Expand "Applications of g-C3N4 Photocatalysts":

While you touch on applications briefly, it would be valuable to have a more in-depth exploration of the practical applications of g-C3N4 photocatalysts, such as their role in environmental remediation and the photodegradation of organic pollutants.

 ·        Integrate "Challenges and Future Directions" Section:

To provide a more comprehensive perspective, I suggest adding a section that outlines the challenges faced in g-C3N4 photocatalysis and discusses potential future directions and avenues for research.

These additions will help strengthen the paper and make it even more informative and valuable to readers. I understand this might require additional work, and I appreciate your efforts in advance.

Best regards

Reviewer 2 Report

Comments and Suggestions for Authors

In this manuscript, Yu et al. reviewed the recent advances in the synthesis and structural design of g-C3N4 materials.

1 1. I suggest the authors to change “graphitized carbon nitride” to “graphitic carbon nitride”, which is a more commonly used term within this area.

2 2. Line 77, the authors talked about the precursors for g-C3N4 synthesis, but only one example from ref.41 is discussed in detail. As a review, other papers with different precursors should also be included to support the claim that “various precursors such as … have been employed…”.

3 3. Add references for Line 78-80.

44. Line 204-226, provide more examples for hard template methods in addition to silica.

55. The authors talked about how different methods influence the surface area, bandgap and .... of g-C3N4 photocatalyst, however, more discussions about the photocatalytic performance are needed. In each section (e.g. 2.1. Precursors and Reaction Temperature; 2.2. C/N Ratio...), the authors should clarify how these factors affect the photocatalytic performance of g-C3N4. For instance, what bandgap width is considered "good" for photocatalytic activity? the narrower the better? What is the relationship between the C/N ratio and the photocatalytic activity of g-C3N4? Are there any related data from photocatalytic studies to support that these strategies are effective?

Comments on the Quality of English Language

Minor editing is needed.

Reviewer 3 Report

Comments and Suggestions for Authors

In the review entitled "Recent Advances in g-C3N4 Photocatalysts: A Review of Reaction Parameters, Structure Design and Exfoliation Methods", Pei and co-workers described several factors contributing to the preparation and utilization as a catalytic material, the g-C3N4 nanoparticles. This is a comprehensive review of the subject and worth publishing in Catalysts. Only minor revision has to be done before publication:

1. In the case of large subsections (3.1 and 3.2), a small summary (e.g., bullet points) of the most important findings should be added at the end of each paragraph.

2. Please add to the Conclusions a paragraph about the best approaches and methods of obtaining high-active g-C3N4-based catalytic materials.

Round 2

Reviewer 1 Report

Comments and Suggestions for Authors

Dear Authors,

Thank you for your contribution to the field. After revision, and changes made by the authors, the manuscript is suitable for publication.

Best regards